# Sensory complexity and global gain in a DCNN codetermine optimal arousal state

**Lynn K.A. Sörensen**
University of Amsterstam
l.k.a.sorensen@uva.nl

**Heleen A. Slagter**
Vrije Universiteit
h.a.slagter@vu.nl

**H. Steven Scholte**
University of Amsterdam
h.s.scholte@uva.nl

**Sander M. Bohté**
Centrum Wiskunde &
Informatica
s.m.bohte@cwi.nl

## Abstract

Arousal deeply impacts behaviour and sensory processing. Whereas for an easy task, perceptual performance linearly increases with higher arousal levels, for a more challenging task, an inverted-U-shaped relationship has been described. These findings are commonly referred to as the Yerkes-Dodson law (1908). Yet, it remains unclear why perceptual performance decays with high levels of arousal in difficult but not in simple tasks. Based on recent studies linking a global gain change in sensory processing to changes in arousal state, we augmented a deep convolutional neural network with a global gain mechanism to mimic the effects of cortical arousal. With this approach, we show that the Yerkes-Dodson law can be accounted for by this global gain mechanism, acting on a hierarchical sensory system that processes stimuli of varying sensory complexity. By leveraging the full observability of our model, we reconcile conflicting findings from previous studies on sensory processing, by showing that both linear as well inverted-U-shaped gain profiles emerge in the interaction of hierarchical sensory processing and global arousal changes.

## 1 Introduction

The effects of neuromodulation are pervasive in the brain. Recent estimates are that around 11% of the variance of spontaneous activity in visual cortex is related to arousal state [1]. For behaviour, already in 1908, Yerkes and Dodson [2] established that an animal's ability to learn a sensory discrimination task was deeply affected by its arousal state (which they manipulated by electrical stimulation). This effect took the shape of an inverted U for challenging perceptual discriminations, indicating that the animals learnt the discrimination best at intermediary levels of stimulation. Interestingly, this relationship became linear for perceptually easy tasks: The animals were fastest at learning when confronted with the strongest stimulation intensities. This interaction between arousal state and perceptual task difficulty and its effect on performance has been replicated many times (e.g. [3]). Yet, the origin of this interaction is much less clear. Some have proposed that an involvement of frontal cortices (e.g. prefrontal cortex) is the decisive factor for an inverted U-shaped performance profile [4], while other suggest that the saturation of activation levels in the locus coeruleus – the central source of noradrenaline – is at the core of this transition [5]. Yet, these broad proposals still do not specify *how* performance degrades with increasing levels of arousal for a perceptually difficult task.

Studies examining the effects of arousal (indexed by pupil dilation) on processing in sensory cortices and perceptual performance provide conflicting answers to this question [6]. For early visual cortex,

2nd Workshop on Shared Visual Representations in Human and Machine Intelligence (SVRHM), NeurIPS 2020.

a number of studies report a linear increase in firing rates with increases in arousal during sensory stimulation [7–10]. One of these studies also reported that the animals performed best at a go/no-go task with perceptual discrimination at the highest levels of arousal, identified through locomotion [8]. Yet, another study in auditory cortex that also used a go/no go-task discovered an inverted U-shaped link for both firing rates during sensory stimulation and performance levels [11]. In contrast to studies in visual cortex, this finding thus suggests an optimum for both a gain in firing rates as well performance at intermediate levels of arousal.

The differences across these studies have been mainly attributed to the difference in sensory modality [6], yet more recent studies suggest a high degree of similarity for the effects of arousal on both visual and auditory processing [12]. Beyond the sensory modality, another striking difference between these studies is the complexity of sensory stimuli that were presented to the animals. While the study with the inverted U-shaped relationship used quite complex sensory stimuli for their go/no go task (presence of pure tones embedded in sequences of structured noise, [11]), the studies suggesting a linear link used rather simple stimuli (presence of a large Gabor patch on a grey screen, [8, 10]). This prompted us to ask whether the different findings of arousal on evoked sensory processing could be due to a difference in experimental stimuli instead.

To answer this question, we implemented a global gain model to mimic the effects of cortical arousal on sensory processing in a deep convolutional neural network (DCNN) and tested it on a range of tasks with varying degrees of sensory complexity. With our approach, we show that long established effects of arousal on perceptual performance, as captured by the Yerkes-Dodson law, can be accounted for by a global gain mechanism acting on a hierarchical sensory system processing stimuli of varying complexity.

## 2 Global gain as a model for cortical arousal

Cortical arousal is linked to increased firing rates in response to sensory stimulation in sensory cortices, i.e. increased gain [6]. These increases in gain have been argued to be implemented via a global release of noradrenaline [13], making them spatially unspecific and temporally slow. For creating an analogous situation in a DCNN, we used a ResNet18 architecture [14] with sigmoidal-like activation functions [15] that was pretrained on ImageNet [16] and achieved a top-1 accuracy of 64.04% on the validation set. We here chose to use a sigmoidal-like function instead of the more commonly used rectified linear unit (ReLU) as our activation function because it more accurately captures the saturating property of biological neurons at extreme values, a characteristic we deemed to be central for studying the effects of gain [17]. To simulate the effects of arousal, we scaled all activation functions in the network simultaneously with one global gain parameter $\gamma$:

$$f(S) = \gamma \max \left( 0, \frac{h}{\exp\left(\frac{c_1 S + c_2}{c_3 S + c_4}\right) - 1} - c_0 + \frac{h}{2} \right)$$

where $f(S)$ describes the outgoing activation and $S$ is the incoming activation. The second part of the equation describes the sigmoidal-like properties of the activation function where the constants $h$, $c_0$, $c_1$, $c_2$, $c_3$ and $c_4$ were derived from a spiking neuron model capturing the mapping between the incoming current and the average post-synaptic potential over infinite time steps (please see [15] for more details and the full derivation from the spiking neuron model). In brief, changing the global gain parameter thus scaled the output of the activation functions, thereby increasing or reducing response gain everywhere in the network.

## 3 Results

### 3.1 Yerkes–Dodson law – A global gain model accounts for behavioural effects of arousal

To evaluate the link between global gain and perceptual difficulty, we developed a task in which we could parametrically alter the visual complexity required to solve a binary discrimination task (Figure 1A). The network was tasked to distinguish a natural scene from an average image. The perceptual difficulty was determined by the number of images that were used to compute the average

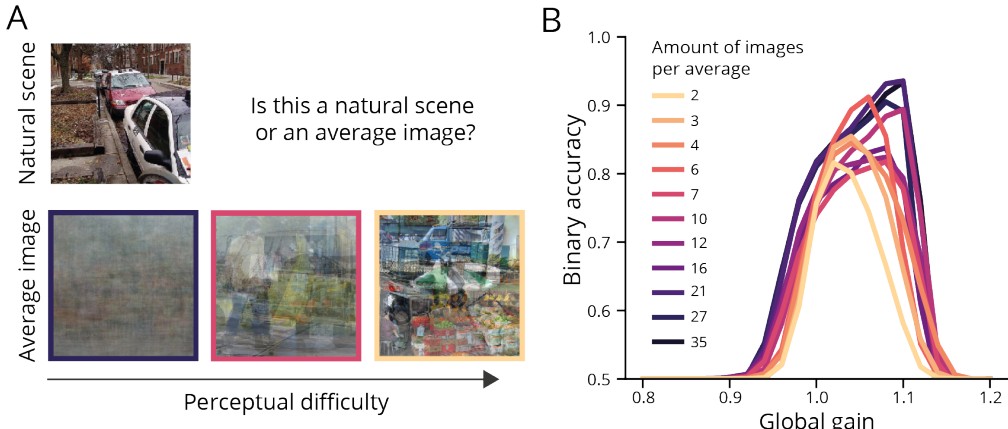

Figure 1: A: Parametric perceptual discrimination task in which the network is tasked to distinguish a natural scene from an average image. The perceptual difficulty decreases with the number of images used for the average image. B: Different ResNet18 models with a single sigmoidal read-out neuron were tested on this task while varying global gain. While we see a clear inverted U-shaped performance profile for the difficult task, the link between global gain and performance becomes more monotonically increasing the easier the task gets.

images, with average images that consisted of fewer images being more difficult to distinguish from the natural scenes. Next, we fine-tuned a single sigmoidal neuron connected to the fully-connected layer of the ResNet18. The remaining network weights were kept unchanged. For all degrees of perceptual difficulty, a separate model was fine-tuned. A single fine-tuned model per perceptual difficulty was then tested with varying degrees of global gain (Figure 1B).

For the most difficult task, based on an average of two images, we observed an inverted U-shape relationship between global gain and performance. The best task performance was achieved close to a gain of 1, corresponding to the evaluation of the original model. Further in- or decreases in gain only hampered performance, thus resulting in an inverted U-shaped profile. Interestingly, the easier the task became the more this inverted U transformed into a monotonically increasing performance profile that had its optimum at very large gain values. This means that the model performed best in a state that was very different from training. Across all perceptual difficulties, performance decreased back to chance performance at a similar global gain value.

This pattern of findings directly replicates the Yerkes-Dodson law [2], assuming that very low and very high levels of global gain are not accessible experimentally or plausible to occur in a living organism. Moreover, we show that the transition from a linear to an inverted U-shape profile is a continuum and could arise directly from applying a global gain factor to a hierarchical sensory system such as a DCNN.

### 3.2 From binary to multi-class tasks - The Yerkes Dodson law also holds for object recognition

A commonality of many studies describing the Yerkes-Dodson law is the usage of sensory discrimination tasks with artificial stimuli that typically involve the detection of a single target class [2, 3, 8, 11]. Yet despite this commonality, these findings have been interpreted to apply to task difficulty more broadly [4, 5]. To test the generalization of our first experiment that the inverted U-shaped performance profile arises in tasks with complex visual stimuli, we therefore reasoned that it should also hold for other visual recognition tasks with more than one target class. To this end, we fine-tuned the same ResNet18 on a 8-way multi-label object detection task with a shared context (Figure 2A).

As in the binary task, we again only trained an output layer consisting of eight sigmoidal output nodes. The networks were first trained with multi-label images, but evaluated on single target images. We find that also here the same pattern of results holds: Across both contexts, we observe an inverted-U-shaped performance profile (Figure 2B). Comparing the datasets shows that networks performed

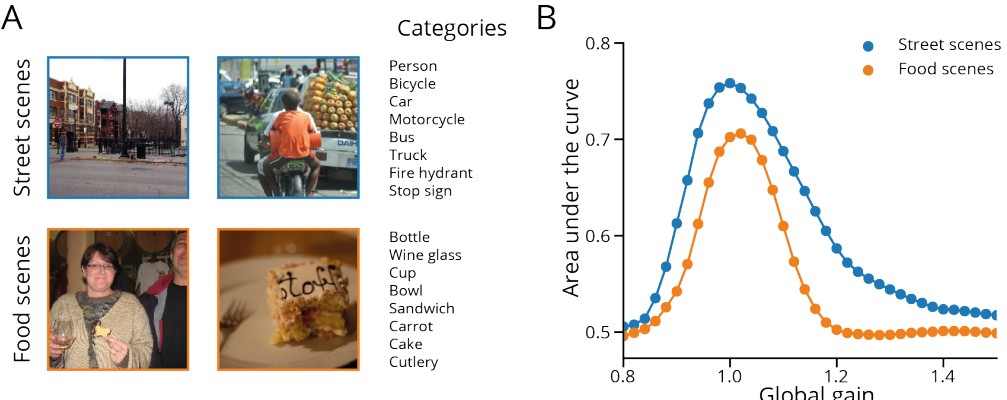

Figure 2: A: Example images from the datasets used to probe the effects of global gain on object recognition. Specifically, we curated two datasets from the MS COCO database [18] with a homogeneous context and 8 possible target categories. Please note that the original images did not feature coloured frames. B: Performance of the finetuned ResNet18 networks for both datasets. Both datasets also result in an inverted U-shaped profile.

less well on the food compared to the street scenes at baseline, a gain of 1. In the data by Yerkes and Dodson [2] more difficult tasks resulted in a narrower inverted U-shape than less difficult task. Similarly in our results, the food scenes featured a narrower profile compared to the street scenes. For high gain regimes, we observed some marked differences to the binary task for the street dataset. While performance completely decayed in the food scenes for global gain parameters exceeding 1.2, street scenes are being solved above chance far beyond that. This could be indicative of a higher chance of guessing based on an overly driven network for the street compared to the food scenes.

In sum, we have thus shown that the Yerkes-Dodson law also holds for more complex behavioural scenarios in our model, such as object recognition in natural scenes. This experiment also further supported our claim that the Yerkes-Dodson law in behaviour is likely moderated by sensory complexity rather than other task factors.

### 3.3 The inverted U arises from high-level features

In a final analysis, we sought to connect our results for performance to observations made experimentally in sensory cortices [6–11]. Specifically, arousal has been linked to both linear as well inverted-U-shaped neural gain profiles for evoked processing. To obtain an intuition on how activation levels were shaped by both stimulus complexity and global gain in our networks, we collected the mean activations of the activation functions after the addition in the ResNet18 across different gain parameters and stimulus complexities.

A general finding across all layers is that the two most perceptually difficult binary task datasets were producing the highest activation levels in almost all ResNet blocks (Figure 3). Furthermore, while there seems to be a linear effect of global gain for most early ResNet blocks $(1-4)$, this link becomes increasingly complex in the later blocks $(5-8)$. Overall, this pattern is suggestive of increasingly specific features in later ResNet blocks, which cannot be activated to a similar extent by simpler visual stimuli such as those used for the perceptually easy task (cf. Figure 1A). Our analysis also makes it clear that both linear as well inverted-U-shaped activation profiles may occur in a hierarchical sensory system. The gain profiles observed experimentally [11, 10] that were first thought to be at odds with one another [6], are likely to co-occur when accounting for stimulus complexity and hierarchical processing.

This last result indicates that the characteristic inverted U-shape observed in perceptually difficult tasks might stem from an interaction between complex sensory features that are tuned to a specific activation range and global gain increases. We speculate that increases in arousal lead to activation patterns that do not match these complex features anymore, thus leading to a decay of these features and hence to lower mean activations at these stages of processing. For behaviour, this interaction between global gain and stimulus complexity can have direct consequences for perceptual performance: If the

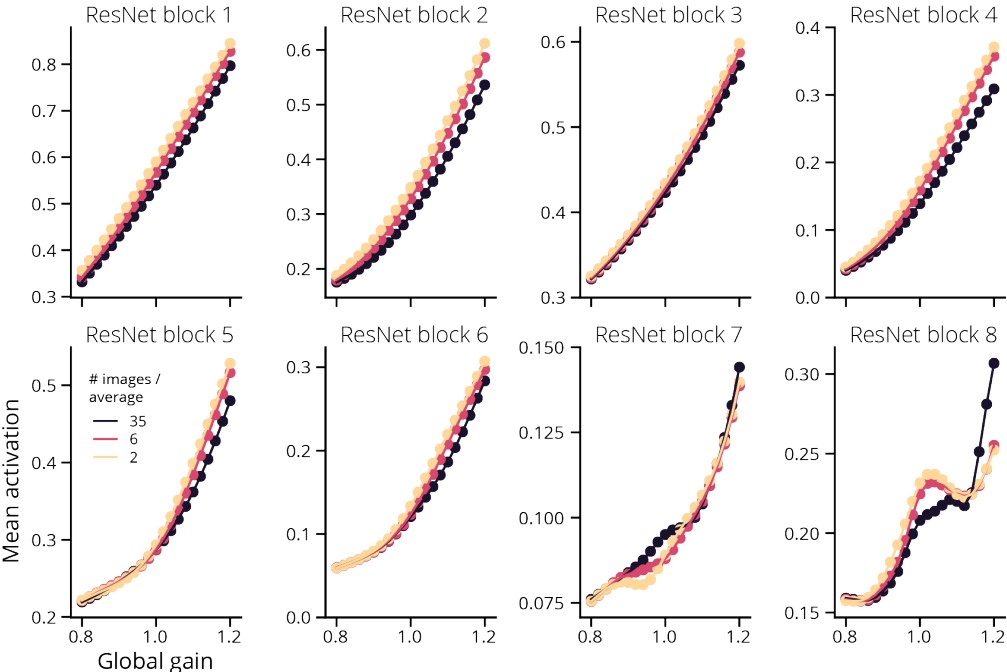

Figure 3: Mean activations of the activation layers after each addition layer in the ResNet18. The used stimuli were the same as in Figure 1.

read-out layer learnt to capitalize on these features during training, this can either result in a linear or an inverted U-shaped performance profile.

# 4   Conclusions

In this paper, we implemented a global gain mechanism in a DCNN to mimic the effects of arousal on sensory processing. Combining performance-optimized computer vision models with recent evidence from neuroscience allowed us to address the long-standing question of why perceptual performance decays with high levels of arousal in perceptually difficult but not in simple tasks. Specifically, we found that with our straightforward global gain model we could robustly replicate the Yerkes-Dodson law for both binary as well as more challenging multi-label object recognition tasks. Beyond that, we have provided an intuition of how the switch from linear to inverted-U shaped performance profiles can arise, namely from an interaction between high-level sensory features and global gain increases. Importantly, we hereby put forward that this often reported change in performance profile for perceptual discriminations is at least in part due to an interaction with sensory processing, and not just the involvement of frontal cortices [4, 5]. Based on our results, we predict that an inverted-U-shaped gain profile with complex sensory stimuli should transition to a more linear gain profile with simpler sensory stimulation, both for perceptual performance as well as sensory activity.

Neuromodulation and arousal in particular have been suggested to belong to different behavioural repertoires in the context of reinforcement learning [5, 19]: While medium levels of arousal are thought to be particularly suitable for exploitative regimes, high levels of arousal were proposed to be particularly suitable for exploratory regimes, thus facilitating the switch to a new task. Our results provide a starting point for addressing these types of questions more directly in a performance-oriented framework with complex sensory processing.

## Acknowledgments and Disclosure of Funding

This work was funded by an NWO Talent Grant (406.17.554) awarded to LKAS, SMB, HAS and HSS.

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
