# OpenReview forum: "Sensory complexity and global gain in a DCNN codetermine optimal arousal state"
_NeurIPS.cc/2020/Workshop/SVRHM — SVRHM@NeurIPS Poster_

### Official Review · AnonReviewer3 · 2020-10-30
**Clever use of CNNs to explore specific and important question/hypothesis**

**Rating:** 7
**Confidence:** 3

**Review:**

This paper highlights a specific, important question in neuroscience with a falsifiable hypothesis, and explores it cleverly in convolutional neural networks. While there is much work to be done to further validate their findings, discover more precisely what is happening in the convolutional networks mechanistically, and eventually in biological brains, that is work far outside the scope of a workshop paper. I thought the task chosen, with the perceptual difficulty based on the number of images averaged, was perfect for the given question and appreciated that the authors additionally showed the same effect using a different task. As a minor nitpick, I found that the presentation  of the plot in Figure 1B actually served to undermine the effect they found - the overlapping curves made it difficult to really see the u-shape vs linear shape without further close inspection.  As a more major nitpick, I was not especially convinced by the results presented in Figure 3 and Section 3.3, in part because I could not quite follow the logic in their interpretation.

---

### Official Review · AnonReviewer1 · 2020-10-30
**Confirming Yerkes Dodson in DCNNs**

**Rating:** 7
**Confidence:** 4

**Review:**

This paper studies the effect of arousal on visual task performance by modeling it in deep convolutional networks. They recover the Yerkes Dodson law using a global gain model in convnets. This is an interesting result and I would recommend acceptance.

I assume the authors leave out the mathematical description of the models due to a lack of space, but including that would help understanding the global gain model that they propose, and also possibly understand analytically why the inverted U shape occurs. In Figure 2, the peak of the two performance curves seems to occur at a global gain of 1 or just under 1, which is curious, suggesting that no arousal is the best condition for the multi label tasks.

---

### Official Review · AnonReviewer2 · 2020-10-30
**A good example of using DCNNs to explain classic neuroscience observations**

**Rating:** 7
**Confidence:** 4

**Review:**

This paper attempts to explain the U-shaped relationship between the arousal level and perceptual performance in the brain using a computational modeling approach and a specific implementation of global gain control in a deep CNN. The results offer an explanation of the classic neuro-scientific observations of arousal-performance profiles conditioned on task difficulty that is purely based on computations in a feed-forward neural network.

My specific suggestions are below:

-Given the importance of the Global Gain theory, it seems appropriate that authors give a formal description/definition of what global gain is. Even after reading the paper I'm left uncertain whether I have understood what global gain is.

-The authors do not explain why the original ReLU activation functions in RN18 were replaced with sigmoid-like activations? Are the same profiles replicable using other activation functions including ReLU?

-What is the performance of RN18 with sigmoid activation on Imagenet?

-line72: do the authors mean that for every combination of complexity and global gain a new model was fine-tuned? Or global gain is changed post-training?

---

### Public Comment · ~Lynn_Katrina_Annika_Sörensen1 · 2020-12-02
**Response to reviewers**

We would like to thank all three reviewers for their attention to detail and thoughtful comments! We were happy to hear that the reviewers appreciated our question and approach and we incorporated all suggested changes in the camera-ready version.

All reviewers commented on the lack of technical specification. We have now included a more elaborate description of the gain implementation, a motivation for using a custom activation function instead of a ReLU, as well as a report of the validation accuracy of the baseline network on the imagenet validation set.

Based on a comment of reviewer 3, we have updated figure 1 to allow for a better visual separation of the different gain profiles.

Reviewer 1 observed that the performance appeared to peak at values below one, yet the street scenes peaked performance at a gain of 1 and the food scenes at a gain of 1.02.

---

### Decision · Program_Chairs · 2020-11-02

Accept (Poster)